# Health Risk Assessment of PM_2.5_ and PM_2.5_-Bound Trace Elements in Thohoyandou, South Africa

**DOI:** 10.3390/ijerph18031359

**Published:** 2021-02-02

**Authors:** Karl Kilbo Edlund, Felicia Killman, Peter Molnár, Johan Boman, Leo Stockfelt, Janine Wichmann

**Affiliations:** 1Department of Occupational and Environmental Medicine, Institute of Medicine, Sahlgrenska Academy, University of Gothenburg, SE-405 30 Göteborg, Sweden; guskilfe@student.gu.se (F.K.); peter.molnar@amm.gu.se (P.M.); leo.stockfelt@amm.gu.se (L.S.); 2Department of Chemistry and Molecular Biology, University of Gothenburg, SE-412 96 Göteborg, Sweden; johan.boman@gu.se; 3School of Health Systems and Public Health, Faculty of Health Sciences, University of Pretoria, Gezina 0031, South Africa; janine.wichmann@up.ac.za

**Keywords:** air pollution, PM_2.5_, health risk assessment, children, South Africa, trace elements

## Abstract

We assessed the health risks of fine particulate matter (PM_2.5_) ambient air pollution and its trace elemental components in a rural South African community. Air pollution is the largest environmental cause of disease and disproportionately affects low- and middle-income countries. PM_2.5_ samples were previously collected, April 2017 to April 2018, and PM_2.5_ mass determined. The filters were analyzed for chemical composition. The United States Environmental Protection Agency’s (US EPA) health risk assessment method was applied. Reference doses were calculated from the World Health Organization (WHO) guidelines, South African National Ambient Air Quality Standards (NAAQS), and US EPA reference concentrations. Despite relatively moderate levels of PM_2.5_ the health risks were substantial, especially for infants and children. The average annual PM_2.5_ concentration was 11 µg/m^3^, which is above WHO guidelines, but below South African NAAQS. Adults were exposed to health risks from PM_2.5_ during May to October, whereas infants and children were exposed to risk throughout the year. Particle-bound nickel posed both non-cancer and cancer risks. We conclude that PM_2.5_ poses health risks in Thohoyandou, despite levels being compliant with yearly South African NAAQS. The results indicate that air quality standards need to be tightened and PM_2.5_ levels lowered in South Africa.

## 1. Introduction

Particulate matter with an aerodynamic diameter less than 2.5 µm (PM_2.5_) in ambient air is the largest environmental cause of morbidity and premature mortality, causing an estimated 2.9 million premature deaths globally in 2017 [1]. Over the last decades, there has emerged substantial epidemiological evidence for an association between PM_2.5_ exposure and all-cause, cardiopulmonary and lung cancer mortality, as well as a variety of non-communicable diseases, such as cardiovascular, respiratory, and metabolic diseases [2,3].

Due to higher exposures, less surveillance and fewer preventative health care services, the burden of disease caused by PM_2.5_ disproportionately affects populations in low- and middle-income countries, and especially vulnerable groups such as pregnant women, infants, and young children [4]. Children and infants have immature physiological system, higher inhalation rate [5,6,7] and larger lung surface area in relation to body weight, which puts them at higher risk to adverse health effects due to ambient air pollution [8]. Additionally, in comparison with adults, children generally have a higher activity level, which increases their inhalation rate, and often spend more time outdoors [9,10].

The air quality guidelines set by the World Health Organization (WHO) in 2005 establish recommended maximum targets of PM_2.5_ concentrations in ambient air at 10 µg/m^3^ for the annual and 25 µg/m^3^ for the 24-h mean [11]. The South African national ambient air quality standards (NAAQS) for PM_2.5_ were established with the Air Quality Act of 2004, with stepwise tightened regulations until 2030 [12]. The current annual and 24-h NAAQS are 20 µg/m^3^ and 40 µg/m^3^, respectively, permitting a maximum of four exceedances of the 24-h mean per year.

It has been previously established that a reduction of PM_2.5_ exposure in South Africa would lead to substantial health benefits [13]. Rural South Africa is currently undergoing an epidemiological transition with a rise in non-communicable diseases [14], which are connected with exposure to ambient air pollution. However, few studies in South Africa have reported on PM_2.5_ levels and no study has yet focused on health effects of ambient PM_2.5_ air pollution in rural South Africa. Lack of air pollution monitoring and scarce health data mean that knowledge of the health risks of air pollution in sub-Saharan Africa is limited [15].

The US Environmental Protection Agency’s (US EPA) human health risk assessment (HRA) module [16] has previously been applied in South Africa to assess health risks of exposure to coarse particle air pollution [17,18] and sulfur dioxide [19], but not of PM_2.5_ or PM_2.5_-bound trace elements. However, this has been done in many other countries [20,21,22,23,24]. The purpose of this study was to use US EPA HRA model to assess the health risks caused by PM_2.5_ and PM_2.5_-bound trace elements in Thohoyandou, a rural South African community.

## 2. Materials and Methods

### 2.1. Study Area and Population

Thohoyandou is a town in the Thulamela municipality in northern South Africa. It is located in Vhembe, a predominantly rural district bordering Zimbabwe and Mozambique with a few interspersed urban settlements. It is also one of the poorest districts in South Africa and has a very young population, with 40 percent of the population in Thohoyandou being less than 20 years old [25]. Thulamela is a mostly agricultural municipality with few industries and no large-scale mining operation [26]. The climate in Thohoyandou is semi-arid to arid and precipitation occurs primarily during the summer [27]. There is no active air quality monitoring station with data available through the South African Air Quality Information System within the district [28].

### 2.2. Sampling and Data Analysis

PM_2.5_ samples were collected for 24 h (from 9 a.m. to 9 a.m.) every third day between 18 April 2017 and 16 April 2018. Duplicate samples were collected every fifteenth day. Gravimetric analysis was used to determine PM_2.5_ mass. The sampling site (22°58.650′ S and 030°26.646′ E) was located nine meters above ground at the University of Venda, close to the center of Thohoyandou but away from major roads. Due to technical issues two PM_2.5_ measurements were missing, for 23 and 26 March 2018. The sampling process has been described in detail by Novela et al. [29].

X-ray fluorescence spectrometry (XRF; using a XEPOS 5, SPECTRO Analytical Instruments GmbH, Kleve, Germany) was used to determine concentrations of all elements between aluminum and uranium in the periodic table. Measurements below the limit of detection (LoD) were registered as LoD divided by √2 [30]. The duplicate sample values were averaged. Precision was estimated using the root mean square method on the duplicate sample values [31]. The significance of concentration differences between cold and dry vs. warm and humid months was assessed using Welch’s unequal variances *t*-test.

Pearson’s correlation analysis was performed for PM_2.5_ and all analyzed trace elements. Microsoft Excel (Microsoft Corporation, Redmonton, WA, USA) was used to calculate air concentrations from the XRF data; all further calculations were performed in MATLAB R2019b (Mathworks, Natick, MA, USA). Ethics approval (reference No. 19/2020) was received from the Ethics Committee at the Faculty of Health Sciences, University of Pretoria, on the 22nd of January 2020.

### 2.3. Health Risk Assessment

We performed HRAs both for PM_2.5_ and separately for the nine detectable elements for which the United States Environmental Protection Agency (US EPA) lists reference concentrations in their Regional Screening Levels (RSL; US EPA 2019). We followed the standardized US EPA human health risk assessment framework [16]. Inhalation was assumed to be the only route of exposure and exposure was considered to be continuous, in order to account for the large degree to which ambient air infiltrates indoor spaces. Equation (1) was used to calculate average daily doses (ADD) from ambient air concentrations:ADD = (C × IR)/BW, (1)
where C signifies concentration (µg/m^3^), IR inhalation rate (m^3^/day), and BW body weight (kg). Exposures were assessed by calculating field average daily doses (FADD) using Equation (1) with the average yearly concentrations measured in Thohoyandou. Inhalation rates and body weights used for the different subpopulations are found in Table 1. American values were used for the reference population in order to be in line with the studies on which the WHO guidelines were based [11,32].

Equation (1) was also used to estimate safe average daily doses (SADD). For PM_2.5_, SADD were calculated both from the yearly WHO guidelines (10 µg/m^3^) and from the yearly South African NAAQS (20 µg/m^3^). For the trace elements, SADD were based on the US EPA reference concentrations (RfC), below which non-cancer health risks are assumed to be low [32]. The SADD based on WHO guidelines and US EPA reference concentrations were calculated using body weight data from an adult American reference population, whereas the SADD based on the South African NAAQS were based on a local body weight data. Separate FADD were calculated for each age-group, using local data when available (see Table 1). We used US EPA recommended values for inhalation rates, since no local data were available. Non-cancer health risks, expressed as unitless hazard quotients (HQ), were then calculated for each exposure using Equation (2):HQ = FADD/SADD.(2)

Out of the detected trace elements, nickel was the only element classified as carcinogenic by the IARC and with an inhalation unit risk (IUR), listed in the "Regional Screening Levels" for inhalation exposure [34,35]. The carcinogenic risk of nickel exposure from PM_2.5_ was estimated using Equation (3), following the method described by the US EPA [36]:CR = C × IUR(3)
where CR signifies an upper bound estimate of the number of excess cancer cases caused by the exposure. We used the IUR listed by the US EPA for nickel refinery dust, which was 2.4 × 10^−4^ [34], as an approximation of the carcinogenic potential of PM_2.5_-bound nickel.

## 3. Results

### 3.1. PM_2.5_ Concentrations

The annual average ambient air PM_2.5_ concentration for the study period was 11 µg/m^3^, which is above the yearly WHO guidelines (10 µg/m^3^), but below the yearly South African NAAQS (20 µg/m^3^). During the study period the 24-h WHO guideline of 25 µg ⁄m^3^ was exceeded in nine samples, but the 24-h South African NAAQS of 40 µg/m^3^ was not exceeded at any time. Six of the nine exceedances of the WHO guidelines took place during the cold and dry months (May–October) and three during the warm and humid (November–April). The average PM_2.5_ levels were significantly higher during the cold and dry than during the warm and humid months (13 compared to 8.3 µg/m^3^, *p* < 0.01). The estimated precision of the PM_2.5_ measurements was 43 percent, where a smaller number indicates higher precision, whereas the precision for the different trace element measurements varied between 33 and 89 percent.

### 3.2. Trace Element Concentrations

Median concentrations of the 17 elemental components detected in Thohoyandou are presented in Table 2 (mean concentrations in Appendix A). Together, the detected trace elements explained 13 percent of the total PM_2.5_ mass. The element found in the highest concentrations in PM_2.5_ was sulfur, followed by silicon, potassium, and iron. These elements, along with titanium, zinc, and bromine, were found in significantly higher levels during the cold and dry months than during the warm and humid. A few elements were found in higher concentrations during the warm and humid months; however, none of these showed a statistically significant difference. In contrast, nickel levels showed high day-to-day variations but remained stable throughout the year (see Appendix A).

### 3.3. Correlation of PM_2.5_ Trace Elements

Table 3 shows the correlations between PM_2.5_ and the trace elements. Iron, silicon, and titanium were highly correlated with each other and, to a lesser degree, with potassium and bromine. Additionally, sulfur, calcium, and zinc were associated with this group of elements, although with somewhat weaker correlations. All of these substances were significantly correlated with changes in the total PM_2.5_ mass.

### 3.4. Health Risk Assessment

The HQ for the population of Thohoyandou was 1.18 or 0.54, using the WHO guidelines and the SA NAAQS as benchmarks, respectively, see Table 4. HQs for the colder and drier months (1.44 or 0.67 using WHO guidelines or SA NAAQS, respectively) were higher than for warm and humid months (0.90 or 0.42). The risks for children and infants were higher than those for adults because of their higher breathing rate in relation to body weight. The HQ for children was 3.29 and 1.52 using the WHO guidelines and SA NAAQS, respectively, as the benchmark, and for infants 3.58 and 1.65.

HQs for the nine trace elements are found in Table 4. Nickel was the only element with HQ above one for adults (HQs at 1.10, 3.06, and 3.33, for adults, children, and infants, respectively). Furthermore, the excess cancer risk of nickel was 3.4 × 10^−6^.

## 4. Discussion

The average yearly PM_2.5_ concentration measured in Thohoyandou (11 µg/m^3^) is slightly above the yearly WHO guidelines (10 µg/m^3^), but below the yearly South African NAAQS (20 µg/m^3^). However, the daily PM_2.5_ limits set by the WHO (25 µg/m^3^) were exceeded in nine out of the 120 measurement days, suggesting that exceedances of the daily limits are common.

The HRA indicates that PM_2.5_ and particle-bound nickel levels recorded in Thohoyandou pose health risks to all age-groups, but especially to younger individuals. The HQ for PM_2.5_ for adults was 1.18, as compared to the WHO benchmarks, which signifies an increased risk compared to internationally accepted guidelines. The HRA indicated substantially higher risks for children (HQ at 3.29) and infants (3.58) and higher during the cold and dry months than during the warm and humid months. The results also indicate that the high nickel content of the PM_2.5_ contributes to both cancer and non-cancer risk that exceed the acceptable standards set by the US EPA.

### 4.1. PM_2.5_ and Trace Element Concentrations

The PM_2.5_ concentrations recorded in Thohoyandou were relatively moderate in comparison to what has been reported elsewhere in South Africa, although previous knowledge of PM_2.5_ levels in rural South Africa is very limited. A population-weighted average PM_2.5_ concentration of 26.6 µg/m^3^ for urban areas in South Africa was calculated by Norman et al. based on data from 2000, with levels ranging from 17.0 µg/m^3^ in City of Cape Town to 39.5 µg/m^3^ in the Vaal Triangle [38]. Morakinyo et al. reported similar PM_2.5_ concentrations from an industrial area in Pretoria (38.2 µg/m^3^ for winter, June and July, and 22.3 µg/m^3^ for summer, January, and February) [39].

The levels recorded in Thohoyandou are more similar to those reported from high-income countries. For example, they are in line with or higher than those recorded for some major urban centers in Europe and higher than all of the seven urban centers in Sweden included in the 2017 EU "Urban PM_2.5_ Atlas" [40]. A much lower seasonal average (5.6 µg/m^3^) was also recorded during winter in a rural community in western Sweden [41].

Particle levels were significantly higher during the cold and dry months than during the warm and humid months. This is in line with many previous studies [39,42,43]. Meteorological factors, domestic fuel burning, and varying agricultural activities may contribute to this difference. The trace element analysis showed that not only levels but also the composition of PM_2.5_ varied throughout the year. We found high correlations between trace elements typically connected to crustal origins (see Table 3), similar to Chimidza and Moloi, who conducted a similar trace element analysis with source apportionment in three locations in eastern Botswana in 2000 [37]. These elements were also highly correlated with total PM_2.5_ mass, indicating that crustal material is an important source of PM_2.5_ in Thohoyandou.

The most common elemental component in the recorded PM_2.5_ was sulfur. The median levels of sulfur recorded in Thohoyandou during the cold and dry season (see Table 2) were between those recorded in different locations by Chimidza and Moloi. Regarding biomass burning and coal combustion as potential sources of sulfur, Chimidza and Moloi concluded that “the most probable source for [sulfur] in Serowe is coal combustion” [37]. In Thohoyandou, however, other indicators of coal combustion, namely chlorine and bromine, were neither as present as in Botswana, nor showed any correlation with sulfur (see Table 3). Additionally, there are no large coal-fired power-plants in the vicinity of Thohoyandou, indicating that biomass burning is instead the most likely source of sulfur in PM_2.5_ in Thohoyandou.

### 4.2. Health Risks

The HQs presented in Table 4 show that despite PM_2.5_ levels in Thohoyandou only exceeded the WHO guidelines slightly and were well below South African NAAQS, HQs were above one, indicating health risks. The lower average body weight of the adult South African population compared to the US American reference population contributed to the higher HQs. Furthermore, as noted above, the levels recorded in Thohoyandou were comparable to or higher than levels recorded in North American and European cities. The substantial health effects reported from these countries suggest that the levels recorded in Thohoyandou could be similarly harmful, despite being moderate in a South African context. It has also been strongly suggested that exposure at concentrations below the WHO guidelines can also negatively impact human health [44,45]. The WHO also stated in their 2013 review of the guidelines that “recent long-term studies show associations between PM and mortality at levels well below the current annual WHO air quality guideline level for PM_2.5_” [46].

Whereas the HRA indicates that the adult population faces health risks only during the cold and dry months, children and infants are exposed to doses much higher than those affecting adults throughout the year. Due to their higher inhalation rate-to-body weight ratio, infants and children face HQs approximately three times those of adults. In addition to this, children are in general exposed to higher levels of soil dust, both since they usually spend more time outdoors and since they are inherently closer to the ground [7,8]. Likewise, there are likely to be other groups with higher vulnerability and susceptibility, e.g., elderly or people with diseases who are at higher risk for the relevant adverse health effects.

Thabethe et al. [18] has previously applied the HRA method in South Africa with special regards to the younger population. However, Thabethe et al. calculated different SADDs for different age groups, resulting in equal HQs. This method implies that the same concentration poses the same risk to adults, children, and infants, even though the latter two are exposed to a higher dose. In this study, HQs were also calculated separately for each age group. However, these were calculated using the same SADD, i.e., under the assumption that the same inhaled amount per body weight would pose the same risk to adults, children, and infants, similar to Traczyk and Gruszecka-Kosowska [24]. Presumably, this is also an underestimation, since the method does not consider that younger individuals have a higher susceptibility and vulnerability.

Under the present South African NAAQS for PM_2.5_ the highest acceptable average PM_2.5_ level will be successively lowered in three steps [12]. The current regulations stipulate a maximum yearly average of 20 µg/m^3^ and will be lowered to 15 µg/m^3^ in 2030. These are both above the current WHO guidelines and the levels we report from Thohoyandou, rendering the regulation non-effective in protecting the health of the population in Thohoyandou. In order to reach levels where the risk posed by exposure to ambient air PM_2.5_ no longer exceeds accepted health risks for adults, children, and infants in Thohoyandou (i.e., a HQ < 1) these regulations would need to be lowered to 9.2 µg/m^3^, 3.3 µg/m^3^, or 3.0 µg/m^3^, respectively. A revision of the WHO guidelines is currently ongoing but not yet published.

For adults, the only trace element with non-cancerous HQ above one was nickel. Nickel from particulate matter has been connected to increased wheeze during the first two years of life [47]. Moreover, PM_2.5_ containing nickel, as well as a variety of other metals (e.g., iron, copper, and zinc), have been shown to increase the prevalence of respiratory infections as well as asthma in both children and adults [48]. Nickel was also the only element with carcinogenic potential documented by the IARC that was found in detectable levels. The excess cancer risk calculated using the method indicated by the US EPA showed that nickel causes an excess of 3.4 cancer cases per million inhabitants per year in Thohoyandou.

Chloride and titanium resulted in HQs above one for children and infants, but not for adults. Exposure to inhaled chlorine can cause respiratory symptoms and disease, e.g., wheeze, cough, dyspnea, shortness of breath, pneumonitis, and abnormalities in diagnostic imaging [49]. Particle-bound titanium can cause negative health effects and have been linked to an increase in biomarkers related to endothelial function as well as increased mortality [50,51].

### 4.3. Strengths and Limitations

Since few studies, including health risk assessments, on the health effects of PM_2.5_ in ambient air pollution have been undertaken in sub-Saharan Africa and regular monitoring is not well-developed [15], this study adds an important insight on the health effects of PM_2.5_ air pollution in rural southern Africa. A strength of this study is that Thohoyandou, as a comparably non-industrialized rural town in a semi-arid climate, with both agricultural activities and some persistence of biological fuel combustion for heating and cooking, is presumably representative of the PM_2.5_ exposure faced by a relatively large and little studied population. This study is limited by using data from only one measuring station and by the relatively low precision of the concentration data. However, the data were collected at a background site with no known local sources of pollution, which would be similar to residential areas at a distance from larger thoroughfares without local fuel combustion.

The HRA methodology is standardized by the US EPA and is, in its simplest form, a way to relate the risk of a certain exposure to the risk associated with a pre-defined acceptable exposure, which in practice is used as a "unit risk". Strengths of the method is that it is cost-efficient, standardized, and requires no health data. However, the HRA method assumes that there is a threshold level below which exposure is risk-free, and that risks associated with exposure above this level increase linearly. The HRA method also does not account for synergies between exposure to different pollutants, variability in the toxicity of a certain exposure, e.g., depending on particle composition, or differences in population sensitivity.

The use of the WHO guidelines and SA NAAQS as benchmarks to establish a “safe” exposure level is both a strength and a limitation. It is problematic insofar that there is no toxicologically established lowest observed adverse effect level for PM_2.5_, and because the guidelines may be seen less as a threshold, below which there may be expected no health effects of exposure, but rather as a target towards which countries may strive in order to improve public health. However, the use of epidemiologically established benchmarks also means that the problem of calculating the effective dose, i.e., the amount of the pollutant that comes into contact with or traverses the respiratory membrane is much lower than the concentration in ambient air, is largely circumvented [52]. Since both the WHO guidelines and the US EPA RfC values are epidemiologically derived these effects are considered from the outset.

A further limitation to this study is the lack of local data on inhalation rates. Instead, we have used standard values for inhalation rate from US EPA "Exposure Factors’ Handbook". However, the recommended inhalation rates are both subdivided according to age group and consider the correlation between body weight and inhalation rates and can therefore be expected to be similar to the inhalation rates of the Thohoyandou population. We used local estimates for body weights, but decided to retain the US estimates for the reference population, since the WHO PM_2.5_ guidelines were developed based on studies conducted in an adult US study population.

Lastly, it is worth noting that the equations we used for calculating the HQ (Equation (2)) are analogous to the ones used by a number of other studies [20,21,22,53], but different from others [17,54]. We define the HQ as the quotient between the actual dose of the exposed subgroup (FADD) and the established safe dose (SADD) calculated from exposure to a benchmark concentration of PM_2.5_ in a standard population. In this way, we obtain a unitless HQ. In contrast, Jan et al. [54] and Morakinyo et al. [17] define their HQ as the quotient of the field dose and the benchmark concentration itself, i.e., a quotient of a dose and a concentration, which, apart from making the HQ unit dependent, bears the risk of understating health risks, with potential implications on public health.

## 5. Conclusions

The results of this study indicate that PM_2.5_ ambient air pollution in Thohoyandou, despite levels being seemingly reassuring in the South African context, could have substantial adverse effects on health of the population in Thohoyandou. The health risk assessment indicates that the general adult population faces health risks only during the cold and dry months, while children and infants are exposed to doses much higher than those affecting adults throughout the entire year. Levels of particle-bound nickel surpassed reference concentrations and the carcinogenic hazard of nickel exposure exceeded the accepted risk level established by the US EPA, with the carcinogenic health risk assessment indicating an upper bound estimate of 3.4 excess cancer cases per million inhabitants. However, it is difficult to point out the source of nickel emissions in Thohoyandou, which highlights the need for further research into the sources of PM_2.5_ and the health effects of the high nickel levels, in addition more immediate measures to reduce nickel emissions affecting Thohoyandou.

We conclude that current South African air quality regulations are not sufficiently effective in protecting the health of the population in Thohoyandou. In order for the regulations to be effective in protecting the health of all age-groups the accepted levels would need to be substantially lowered. The rise in non-communicable diseases, which are generally exacerbated by exposure to air pollution, in rural South Africa as part of the epidemiological transition, makes it probable that the health impact caused by air pollution will increase.

## Figures and Tables

**Table 1 ijerph-18-01359-t001:** Variables and assumptions used for the health risk assessment.

Variable	Population	Value	Source
Body weight	Adults (Limpopo)	68.1 kg	US AID [33]
	Children (South Africa)	13.8 kg	US AID [33]
	Infants (South Africa)	7.6 kg	US AID [33]
	Reference	73.7 kg	Ogden et al. [32]
Inhalation rate	Adults and reference	14.9 m^3^/day	US EPA [9]
	Children	9.0 m^3^/day	US EPA [9]
	Infants	5.4 m^3^/day	US EPA [9]

**Table 2 ijerph-18-01359-t002:** Median trace elemental concentrations in ambient PM_2.5_ in Thohoyandou, for the whole year and separately for warm and humid months (November to April) and cold and dry months (May to October). Results from Chimidza and Moloi [37], who conducted their sampling during winter (June to July), are included for comparison. All figures are in ng/m^3^.

	Thohoyandou					Chimidza and Moloi, 2000		
	Annual Median	St. dev.	Nov–Apr Median	May–Oct Median	*p*-Value	Median (Serowe)	Median (Selibe-Phikwe)	Median (Francis-Town)
PM_2.5_	8000	8300	6300	11,000	0.007	-	-	-
Si	160	300	120	220	<0.001	*,^1^	*,^1^	*,^1^
S	350	650	270	460	<0.001	1100	280	320
Cl	22	120	21	24	0.911	60	63	62
K	100	220	62	180	<0.001	390	250	290
Ca	65	82	53	75	0.030	170	67	44
Ti	27	27	22	37	<0.001	15	18	7.1
V	0.64	10	0.64	0.64	0.727	*	*	*
Mn	3.90	4.4	4.7	3.2	0.276	6.3	5.1	2.6
Fe	110	120	80	140	<0.001	210	260	100
Ni	14	5.0	14	14	0.924	1.7	2.3	0.6
Cu	2.3	5.2	2.3	2.2	0.972	2.3	2.3	1.1
Zn	1.9	3.2	1.0	2.9	0.002	6.8	3.4	3.7
Br	3.7	4.1	2.7	5.2	<0.001	8.0	8.5	8.3
Sr	2.2	1.3	2.4	2.0	0.853	1.1	0.6	0.3
Sb	25	18	27	24	0.877	-	-	-
Ba	20	11	21	20	0.310	-	-	-
U	2.5	2.3	2.6	2.4	0.091	-	-	-

-, not measured; *, below limit of detection (LoD) (elements below LoD in Thohoyandou not shown); ^1^, Chimidza and Moloi have a LoD for silicon at 720 ng/m^3^.

**Table 3 ijerph-18-01359-t003:** Pearson’s correlation coefficients for PM_2.5_ and detected trace elements. Correlations significant at α = 0.05 level after Holm–Bonferroni correction are in bold figures.

	PM_2.5_	Si	S	Cl	K	Ca	Ti	V	Mn	Fe	Ni	Cu	Zn	Br	Sr	Sb	Ba	U
PM_2.5_	1.0	**0.6**	**0.6**	0.0	**0.6**	**0.3**	**0.6**	−0.1	−0.2	**0.6**	0.1	0.0	**0.4**	**0.6**	0.0	0.1	−0.3	0.0
Si		1.0	**0.7**	0.1	**0.8**	**0.6**	**0.9**	−0.1	−0.2	**0.9**	0.0	0.0	**0.5**	**0.6**	0.0	0.1	0.0	0.0
S			1.0	0.0	**0.5**	0.2	**0.6**	−0.1	−0.2	**0.6**	−0.2	−0.2	**0.4**	**0.6**	0.0	0.1	−0.1	0.0
Cl				1.0	0.0	0.3	0.1	0.0	−0.1	0.1	−0.2	0.1	0.1	0.1	0.0	0.0	0.0	−0.1
K					1.0	**0.5**	**0.8**	−0.1	−0.2	**0.8**	0.0	−0.1	**0.4**	**0.8**	0.1	0.1	0.0	0.0
Ca						1.0	**0.7**	−0.1	0.0	**0.6**	0.1	0.2	**0.5**	**0.4**	0.1	0.1	0.2	0.0
Ti							1.0	−0.1	−0.2	**0.9**	0.0	0.0	**0.5**	**0.6**	0.1	0.2	0.1	0.1
V								1.0	0.0	−0.1	0.0	0.0	−0.1	0.0	0.0	−0.1	0.0	0.0
Mn									1.0	−0.2	0.1	0.2	0.0	−0.3	−0.1	0.1	0.1	0.0
Fe										1.0	0.0	−0.1	**0.5**	**0.7**	0.0	0.1	0.0	0.0
Ni											1.0	0.2	0.1	−0.1	0.2	0.1	−0.1	0.1
Cu												1.0	**0.5**	−0.1	0.1	0.1	0.0	0.1
Zn													1.0	**0.4**	0.1	0.1	0.0	0.0
Br														1.0	0.0	0.1	−0.1	−0.1
Sr															1.0	0.1	0.0	**0.4**
Sb																1.0	0.0	0.2
Ba																	1.0	0.0
U																		1.0

**Table 4 ijerph-18-01359-t004:** Benchmark concentrations and hazard quotients (HQs) for PM_2.5_ and the detected trace elements for which US EPA gives reference concentrations. HQs ≥ 1, indicating a health risk, are in bold figures.

	Benchmark (µg/m^3^)	Adults			Children			Infants		
		Annual	Nov–Apr	May–Oct	Annual	Nov–Apr	May–Oct	Annual	Nov–Apr	May–Oct
PM_2.5_	20 *	0.54	0.42	0.67	**1.52**	**1.16**	**1.87**	**1.65**	**1.27**	**2.04**
PM_2.5_	10 **	**1.18**	0.90	**1.44**	**3.29**	**2.51**	**4.03**	**3.58**	**2.74**	**4.39**
Si	3	0.09	0.06	0.12	0.26	0.18	0.34	0.28	0.20	0.37
Cl	0.15	0.44	0.53	0.35	**1.23**	**1.48**	0.97	**1.33**	**1.62**	**1.05**
Ti	0.1	0.37	0.27	0.47	**1.03**	0.77	**1.30**	**1.13**	0.83	**1.42**
V	0.1	0.05	0.05	0.05	0.14	0.13	0.14	0.15	0.14	0.16
Mn	0.05	0.11	0.12	0.09	0.30	0.34	0.26	0.33	0.37	0.28
Ni	0.014	**1.10**	**1.13**	**1.06**	**3.06**	**3.15**	**2.97**	**3.33**	**3.43**	**3.23**
Sb_2_O_3_	0.2	0.10	0.09	0.10	0.27	0.26	0.28	0.29	0.28	0.30
Ba	0.5	0.04	0.04	0.04	0.12	0.12	0.12	0.13	0.13	0.13
U	0.04	0.08	0.09	0.07	0.22	0.24	0.20	0.24	0.26	0.22

*, yearly South African national ambient air quality standards (NAAQS); **, yearly WHO air quality guidelines.

## Data Availability

Data are available upon request, but if used in another manuscript the authors wish to be included as co-authors.

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
