# Peer review of "Health Risk Assessment of PM2.5 and PM2.5-Bound Trace Elements in Thohoyandou, South Africa"

_ijerph, 2021, doi:10.3390/ijerph18031359_

Round 1

Reviewer 1 Report

It is a very important issue to diagnose the air pollution and to assess health risk by measuring fine dust for environmental research and public health. From this point of view, it is of great significance both academically and practically that the authors assess the health risk caused by PM2.5 and PM2.5-bound trace elements in a specific area.

However, the authors need to improve the following points. If the authors accept them, I believe that this paper will be a good paper that meets the requirements of IJERPH.

First, the authors need to rewrite the introduction and conclusions more clearly for the readers of this journal.

Second, the authors need to emphasize in more detail the theoretical and practical contributions of this study.

Third, the authors need to add the recent studies about this topic. Some references are so old.

Reviewer 2 Report

A. Abstract:

i) Please remove the following sentence from the abstract; "However, no previous study has assessed the health effects of particulate air pollution in rural South Africa."

ii) Include data type and sources with data collection timeline.

iii) Include a line related to policy suggestions (either govt. or overall).

B. Discussions and Conclusions:

i) Add existing air pollution reduction-related policies and strategies, if any in South Africa.

ii) Discuss separately on potential measures taken by other countries linking South Africa's potential policy measures. Specifically, the double standard of air pollution (WHO and South Africa). Can you suggest step-by-step policy measures?

iii) The exposure for adult and children are sometimes confusing. Need more explanation and supporting references in South African context.

Round 2

Reviewer 1 Report

This paper is well written in terms of the structural and technical level. In addition, the authors faithfully reflected the reviewer's requirements in the revised paper. I believe this paper is eligible for the publication in IJERPH.